# Association between Influenza Vaccination and the Risk of Bell’s Palsy in the Korean Elderly

**DOI:** 10.3390/vaccines9070746

**Published:** 2021-07-06

**Authors:** Nayoung Jeong, Yejee Kim, Chungjong Kim, Sangmin Park, Joongyub Lee, Namkyong Choi

**Affiliations:** 1Department of Health Convergence, Ewha Womans University, Seoul 03760, Korea; jny4311@gmail.com; 2Department of Clinical Epidemiology and Biostatistics, Asan Medical Center, University of Ulsan College of Medicine, Seoul 05505, Korea; kimyejee@amc.seoul.kr; 3Department of Internal Medicine, Ewha Womans University Seoul Hospital, Seoul 07804, Korea; erinus00@gmail.com; 4Department of Family Medicine, Seoul National University Hospital, Seoul 03080, Korea; smpark3.snuh@gmail.com; 5Department of Biomedical Sciences, Seoul National University Graduate School, Seoul National University College of Medicine, Seoul 03080, Korea; 6Department of Preventive Medicine, Seoul National University College of Medicine, Seoul 03080, Korea; tp240@naver.com

**Keywords:** influenza vaccine, Bell’s palsy, elderly people, adverse events, large-linked database, self-controlled risk interval design

## Abstract

Previous studies have shown controversial results on the risk of Bell’s palsy after influenza vaccination. Since the antigenic components of influenza vaccine can vary from season to season, continuous safety monitoring is required. The aim of the present study was to determine whether there was an increased risk of Bell’s palsy in the elderly after influenza vaccination between the 2015/2016 and 2017/2018 flu seasons. This study included the elderly who received influenza vaccinations for three flu seasons using a large-linked database of vaccination registration data from the Korea Disease Control and Prevention Agency and the National Health Insurance Service claims data. We used a self-controlled risk interval design with a risk interval of 1 to 42 days and a control interval of 57 to 98 days postvaccination and calculated the incidence rate ratio. To ensure the robustness of the results, sensitivity analyses were also carried out with different risk and control intervals. Of 4,653,440 elderly people who received the influenza vaccine, there was no statistically significant increase in the risk of Bell’s palsy (IRR: 0.99, 95% CI: 0.92–1.07). Similar results were found in analysis results for each season and the results of the sensitivity analyses excluding the 2017/2018 season. In conclusion, we found no evidence of an increased risk of Bell’s palsy after influenza vaccination. The results of our study provide reassurance about the safety of the influenza vaccine NIP program. However, it is necessary to continuously monitor the risk of Bell’s palsy during future flu seasons.

## 1. Introduction

Influenza, commonly known as flu, is an acute pyrogenic disease that is common and highly contagious [1]. In most cases, healthy people can recover within a few days after infection. However, groups at high risk of influenza infections, such as the elderly and infants, can have complications such as pneumonia and exacerbation of chronic diseases [2]. One of the most effective ways to prevent influenza is vaccination [3]. In Korea, the government has guaranteed a certain level of essential vaccination, including influenza vaccines, to recommended targets through the National Immunization Program (NIP). For those aged 65 years or older, influenza NIP has been conducted since 1997 in public health centers. It has been expanded to private medical institutions since 2015 [4]. According to the ‘Organisation for Economic Co-operation and Development (OECD) Health Statistics’, influenza vaccination coverage among the elderly in Korea was about 83% as of 2017, which was higher than other OECD countries such as the United States (69.1%), Japan (51.0%), and Germany (34.8%) [5].

Bell’s palsy is an idiopathic facial nerve paralysis affecting the cranial nerves. It occurs in 15 to 50 per 100,000 people per year [6,7,8,9]. Bell’s palsy is generally believed to be caused by inflammation of the facial nerves and swelling after infection with viruses such as herpes simplex [10,11,12]. It may also be associated with autoimmune disease, Guillain-Barré syndrome, sarcoidosis, multiple sclerosis, and otitis media [8,13,14]. It is difficult to understand the biological mechanism of Bell’s palsy caused by influenza vaccination. However, several prior studies using large-linked databases have shown the possible association between influenza vaccination and Bell’s palsy [15,16,17].

However, the results of the previous studies on the possible association between influenza vaccination and Bell’s palsy are diverse and still remain uncertain because of differences in the type of influenza vaccine, flu seasons, study design, and data sources. Moreover, while most large database studies to date have been conducted in United States and Europe, there is a lack of evidence on the association between influenza vaccination and Bell’s palsy in Asian populations, including Korea. Additionally, among various age groups, the need for evidence about the elderly is high because the elderly not only receive influenza vaccination through the NIP in Korea, but also show a high risk of Bell’s palsy [8,18]. Thus, the objective of the present study was to determine whether there was an increased risk of Bell’s palsy for the elderly after influenza vaccination between the 2015/2016 season and the 2017/2018 season.

## 2. Materials and Methods

### 2.1. Data Sources

Two national databases, the vaccination registration database from the Korea Disease Control and Prevention Agency (KDCA) and the national claims database from the National Health Insurance Service (NHIS) were used. Computerized registration database of National Immunization Registry Integration System is an integrated database. It was launched to manage vaccination records by National Immunization Program (NIP) in KDCA in 2002 [19]. Vaccination data from public health centers and private medical institutions should be recorded electronically because NIP vaccination is essential for reimbursing the cost of vaccines [19]. The vaccination registration database includes demographic characteristics of vaccinees, information about vaccines provided, and information about medical institutions. Since the National Health Insurance Act was enacted in 1999, the Health Insurance and Review Assessment (HIRA) and the NHIS were established to manage the medical insurance program for most citizens in Korea [20]. The HIRA reviews medical fees for reimbursement decisions and the NHIS reimburses healthcare services in compliance with the HIRA’s decisions. In addition, NHIS reviews the eligibility of insurance policyholders, imposes and collects contributions, and consults with healthcare providers on medical fee schedules [21]. Based on claims data from the reimbursement process, the HIRA and the NHIS have constructed a large healthcare database that can be used for research purposes.

We asked the KCDA and the NHIS to link vaccination registration data and NHIS claims data from 2015 to 2018. First of all, the KCDA delivered vaccination registration data to the NHIS, and the NHIS linked it using resident registration numbers. We were only able to receive and use deidentified linked data. We covered three seasons of influenza vaccination as follows: the 2015/2016 season, corresponding to the period from September 2015 to April 2016; the 2016/2017 season, corresponding to the period from September 2016 to April 2017; and the 2017/2018 season, corresponding to the period from September 2017 to April 2018.

### 2.2. Study Population

In this study, those who had at least one vaccination record from 2015/2016 to 2017/2018 flu season, and those who were aged 65 years or older on the date of influenza vaccination were used as the study cohort. Since NIP of influenza vaccination includes the elderly aged 65 years or older, we excluded individuals who were less than 65 years old because their vaccination record might not be reported necessarily. The elderly might have received influenza vaccinations in multiple flu seasons during the study period, in which case doses of influenza vaccine given in different flu seasons were considered as independent exposures. If an elderly patient received two or more influenza vaccinations during the same flu season, only the first dose of influenza vaccine per season was included in the analysis.

### 2.3. Outcomes

We identified patients who were diagnosed with Bell’s palsy in outpatient and inpatient settings using International Classification of Diseases 10th revision (ICD-10) code G51.0 (Bell’s palsy) and who were prescribed with glucocorticosteroids within 14 days of the diagnosis date using WHO-ATC code H02AB. Among identified cases, we ruled out those who had another Bell’s palsy or diagnosis of related diseases, such as postherpetic geniculate ganglionitis (ICD-10 code B02.21), clonic hemifacial spasm (G51.3), facial myokymia (G51.4), other disorders of facial nerve (G51.8), acute myocardial infarction (I21), nontraumatic subarachnoid hemorrhage (I60), nontraumatic intracerebral hemorrhage (I61), other and unspecified nontraumatic intracranial hemorrhage (I62), cerebral infarction (I63), injury of facial nerve (S04.5), and intracranial injury (S06) recorded within 180 days prior to vaccination to exclude follow-up visits for Bell’s palsy and related diseases.

### 2.4. Study Design and Statistical Analysis

We used a self-controlled risk interval (SCRI) design [22] that used only vaccinated people to compare the risk of Bell’s palsy onset following influenza vaccination in a risk period with the risk in a control period. SCRI design is efficient because only vaccinated cases are informative in this design [23]. Unlike study designs that start with cases such as self-controlled case series (SCCS) [24] and case-crossover design [25] among self-controlled designs, SCRI starts with exposure (i.e., vaccination). Based on this, the risk and control intervals are determined. The SCRI study design allowed us to adjust for time-invariant confounders, such as gender and chronic causes for diseases. In addition, it is based only on those who have experienced exposure, i.e., those who have been vaccinated, making it possible to minimize misclassification bias of such vaccinated cases classified as unvaccinated ones [22].

The null hypothesis of this study using the SCRI study design was that the average risk of Bell’s palsy in the risk interval for the elderly who received influenza vaccine was equal to the average risk in the control interval. We did not select a control period prior to vaccination to avoid underestimating the background rate of Bell’s palsy because the occurrence of Bell’s palsy a few days before influenza vaccination could affect whether patients were vaccinated. We calculated incidence rate ratio (IRR) of Bell’s palsy during the risk interval versus the control interval by fitting a conditional Poisson regression model. In this process, the number of onset cases and person-days in the risk and control intervals was calculated. If vaccinees received two or more doses during the study period, they contributed more person-time than people who received only one dose. We calculated the incidence rate with 95% confidence intervals (CIs) for each interval using the number of onset cases divided by person-time.

As a primary analysis, we used a risk interval of 1–42 days following influenza vaccination and a control interval of 57–98 days following influenza vaccination, referring to previous case reports of Bell’s palsy after receiving vaccination and studies using large databases conducted in other countries (Figure 1). In addition, sensitivity analyses were performed to ensure the robustness of study results by selecting other control intervals. We used risk intervals of 1–14 days and 1–28 days following the vaccination and control intervals of 29–42 days and 43–70 days following influenza vaccination for sensitivity analyses. We set a 14-day washout period between risk and control intervals in all analyses.

We performed subgroup analysis to investigate differences by age group, gender, the month of influenza vaccination, and the history of chronic conditions prior to vaccination. For chronic conditions, we selected diabetes, dyslipidemia, and hypertension, known to be related to the occurrence of Bell’s palsy in previous studies [8,18,26,27]. Subgroups with a history of chronic diseases were established for the elderly who were diagnosed with diabetes, dyslipidemia, and hypertension at least twice during the 6 months prior to vaccination.

The SCRI analysis was conducted using SAS enterprise guide version 7.1 (SAS Inc., Cary, NC, USA) with an alpha level of 5%.

## 3. Results

We identified 4,653,440 elderly patients who received influenza vaccinations during the study period. A total of 11,656,966 doses of influenza vaccine were administered to the study population (Table 1). The number of the elderly patients who met our inclusion criteria and received at least one influenza vaccine during the seasonal period was 3,686,809 in the 2015/2016 season, 3,872,631 in the 2016/2017 season, and 4,097,526 in the 2017/2018 season. This number increased as the season passed. More than 95% of the elderly received the vaccine in October in the 2015/2016 season and 2016/2017 season (Figure 2). However, in the 2017/2018 season, vaccination cases were high at about 36% in September, which was believed to be because the NIP of the influenza vaccine for elderly began on September 26, unlike the other seasons that began in October.

Figure 3 shows the distribution of Bell’s palsy incidence within 98 days postvaccination. The number of Bell’s palsy cases observed per day was between 20 and 50 for all seasons. Seasonal distribution showed that the onset of Bell’s palsy in a short period in the 2015/2016 season was relatively less than those in other seasons. Another feature was irregular peak value, with a large increase in the number of occurrences at 98 days after vaccination in the 2017/2018 season.

The total number of Bell’s palsy cases was determined using three lengths of risk intervals and control intervals. As a result, 1348 cases (1–42 days postvaccination) within the risk interval and 1362 cases (57–98 days postvaccination) within the control interval were identified. The baseline characteristics of cases were similar between the risk and control intervals (Table 2). More than half of cases occurred in those aged between 65 and 74 years. The proportion of women was high in all length intervals. Most patients in the risk and control intervals were those who received the vaccination between September and October. Clinical setting for those who were diagnosed with Bell’s palsy in risk and control intervals after vaccination showed that hospitalization accounted for approximately 20% among all intervals. For sensitivity analyses, 475 cases (1–14 days postvaccination) and 907 cases (1–28 days postvaccination) within the risk interval and 441 cases (29–42 days postvaccination) and 866 cases (43–70 days postvaccination) within the control interval were observed (Appendix A). The distributions of characteristics were similar to those included in the primary analysis.

The risk of onset of Bell’s palsy during the risk interval was not significantly different from the risk during the control interval for the total cases of the three flu seasons (IRR: 0.99, 95% CI: 0.92–1.07) (Table 3). The number of onsets of Bell’s palsy per season increased over time, just as the number of doses per season also increased. For each season, there was no statistically significant risk of Bell’s palsy. Similarly, in most sensitivity analyses, no statistically significant association between influenza vaccination and the risk of Bell’s palsy was observed. However, the IRR in the 2017/2018 season was significantly high (IRR: 1.18, 95% CI: 1.01–1.38) when the risk interval was 1–28 days postvaccination (Table 4).

The risk of Bell’s palsy after influenza vaccination was consistent across subgroups stratified by baseline characteristics: age groups (65 to 74 years of age, 75 to 84 years of age, and over 85 years of age), gender, month vaccinated (September, October, and November), and comorbidities (diabetes, dyslipidemia, and hypertension) (Figure 4). Even for the risk of each subgroup according to the vaccinated month by season, no statistically significant difference was found among those who received vaccination in a specific month (Appendix A). For subgroup analyses, the risk of Bell’s palsy during 1–14 days postvaccination among patients with a history of diabetes (IRR: 1.34, 95% CI: 1.05–1.72) or dyslipidemia (IRR: 1.36, 95% CI: 1.01–1.83) was statistically increased, while there was no risk when the risk interval was 1–28 days or 1–42 days postvaccination (Appendix A).

## 4. Discussion

This study was conducted using a large elderly population in Korea to determine whether there was an increased risk of Bell’s palsy after influenza vaccination. During surveillance in three flu seasons using a self-controlled design, we found no evidence of an increased risk of Bell’s palsy after influenza vaccination in the elderly. In the sensitivity analysis, the risk of Bell’s palsy during 1–28 days following influenza vaccination was high in the 2017/2018 season. However, none of the other analyses showed statistically significant results. Furthermore, there was no significant difference in risk according to age, gender, or month vaccinated, although the risk of Bell’s palsy within 14 days after vaccination was significantly increased in patients diagnosed with diabetes prior to vaccination. To the best of our knowledge, this is the first study in Korea to study the association between influenza vaccination and Bell’s palsy in the elderly using a large-linked database.

The lack of association between influenza vaccination and Bell’s palsy during the three seasons in this study is similar to findings of previous observational database studies. Stowe et al. studied the relationship between Bell’s palsy and inactivated influenza vaccines using an SCCS design [28]. They found that the relative incidence was 0.92 (95% CI: 0.78–1.08), indicating no significant risk [28]. Furthermore, analysis of the association between trivalent inactivated influenza vaccine (IIV) and Bell’s palsy in a U.S. population of those aged under 18 years showed no significant increase in any risk interval [9]. Conversely, some prior studies applying an SCRI design observed significant increases in the risk of Bell’s palsy after influenza vaccination. An analysis of the safety of H1N1 influenza vaccination in the U.S. Sentinel’s Post-Licensure Rapid Immunization Safety Monitoring (PRISM) system showed a significant increased risk of Bell’s palsy in those aged more than 25 years who were vaccinated with only H1N1 vaccine (IRR = 1.65, 99% CI: 1.03–2.64) [16]. In U.S. Vaccine Safety Datalink (VSD), the relative risk of Bell’s palsy after quadrivalent IIV was significantly higher in the 2014/2015 season for those aged 50 years or more [17]. However, additional medical chart review in both studies proved false signals for reasons such as having a history of the disease, having occurred prior to vaccination, or being misrecorded. These prior findings were consistent with our conclusion from the evidence about the safety of the influenza vaccine for the elderly in Korea. However, attention should be paid to the risk in the 2017/2018 season when the risk interval was 1–28 days. It was unclear whether the vaccine strains in that season were actually related to the onset of Bell’s palsy.

Influenza virus strains are recommended by the World Health Organization each year according to influenza viruses expected in the next flu season. During the 2015/2016 season, 2016/2017 season, and 2017/2018 season selected for this study, at least one of three or four virus strains changed every year. In particular, none of the four recommended strains overlapped during the 2015/2016 and 2017/2018 seasons. For this reason, the risk of Bell’s palsy after influenza vaccination might have been especially different for both seasons when the risk interval was 1–28 days in this study.

It is known that the likelihood of Bell’s palsy increases during the cold season based on previously published studies [29,30]. There have been several hypotheses such as reactivations of herpes virus type 1 infection caused by seasonal affective stress [31] or by partial ischemia and inflammation due to vasomotor changes of facial regions exposed to lows temperature [32]. The risk and control sections of the three influenza seasons included in this study included autumn or winter with slight differences in temperature for each season. The risk and control intervals of the three flu seasons included in this study included autumn and winter with slight variations in temperature from season to season. The average temperature in November in the 2015/2016 season was 10.1 degrees Celsius, which was higher than those in other seasons [33]. For this reason, the IRR might have been lower for 1 to 28 days and 1 to 42 days when the risk interval was likely to be included in November. Additionally, in the 2017/2018 season, the average temperature between November and December was about 3 to 4 degrees Celsius lower than those in other seasons, possibly resulting in a high number of Bell’s palsy cases in the risk interval.

In this study, patients with diabetes and dyslipidemia had a significantly higher risk of Bell’s palsy at 1–14 days postvaccination. Given this result, the inclusion of patients with chronic diseases might have affected the study results when the risk interval was set to be 1–14 days. However, since no significant risk of Bell’s palsy was observed in diabetic or dyslipidemic patients when using other risk windows, it was difficult to consider that there was a significant association between chronic conditions and the risk of developing Bell’s palsy after vaccination. To date, among the chronic conditions considered to be related to Bell’s palsy, there are few studies on the association between Bell’s palsy and influenza vaccination. Therefore, it seems necessary to continuously check the risk of adverse events postvaccination in patients with highly relevant diseases such as chronic conditions.

Since influenza vaccination coverage in the elderly population in Korea is high, it is difficult to select appropriate non-vaccinated controls. Therefore, when conducting safety studies of drugs such as vaccines with high exposure rates, a study design that uses the patients themselves as a control group is appropriate. Note that SCRI design has been widely used in active vaccine surveillance systems to screen for the risk of vaccine adverse events because it can be applied relatively simply compared to other self-controlled studies such as SCCS. Some previous studies that used the SCRI design have selected the period prior to vaccination as a control interval. However, it is important to note that vaccination is performed when the subject is the healthiest and that the vaccination is sometimes withheld when the subject is in a poor health condition [34]. Thus, the days immediately preceding the vaccination are likely to be much healthier than usual, which might lead to underestimation of the actual incidence of events if they are included in the control interval. Considering these ‘healthy vaccinee effect’ issues, it is recommended to set up a period that minimizes the effect when selecting a control interval prior to vaccination in a vaccine safety study [22]. Alternatively, the control interval after vaccination might be used to avoid the healthy vaccinee effect as in this study. This effect might also result in a lower incidence of events for a short period of time immediately before and after vaccination [35]. Using the SCRI design, it is difficult to know the exact time period at which the risk of onset of adverse events increases after vaccination. To control the uncertainty in setting risk and control intervals, we conducted sensitivity analyses with shorter or longer intervals. As a consequence, results of this study provided reassurance that we did not miss an increased risk by incorrectly specifying the risk interval.

The current study had several strengths. First, this study targeted a large population using the national vaccination registration database and health insurance database. For this reason, this study had the power to detect relatively rare adverse events following vaccination and the results of this study could be generalized to the Korean elderly population. Second, the SCRI design can be used to avoid exposure misclassification bias by including only vaccinated individuals. In addition, influenza vaccination coverage in Korea is very high in the elderly. Thus, the characteristics of the non-vaccinated group might be quite different from those of the vaccinated group, which might cause a selective bias. We minimized this bias by adopting a self-controlled design without setting the control group as a non-vaccinated group. Time-invariant confounders could be also adjusted because vaccinees placed themselves in the control group. Finally, we conducted several sensitivity analyses using multiple risk and control intervals to ensure the robustness of our primary results.

Along with these strengths, our study also had several limitations. First of all, there were problems with the validity and misclassification bias of the Bell’s palsy diagnostic code when using the claims data. In other words, patients might not have been treated for Bell’s palsy even if they were given the Bell’s palsy diagnostic code. It was possible that they actually suffered from other diseases. To this end, it was necessary to review medical charts to confirm whether the disease occurred in reality. However, in this study, such confirmation could not be performed due to limitations of the data. Second, the data used in this study did not include information on the number of viruses or the type of vaccine. Thus, they could not be analyzed separately. Even with the same influenza vaccine, the risk of adverse events might differ depending on the type of vaccine. Some previous studies [15,16] have shown that there is a difference in the risk of adverse events depending on the number of viruses or the type of influenza vaccines. The influenza vaccine for the NIP program provided to the elderly in Korea is an inactivated trivalent influenza vaccine. We assumed that all subjects were vaccinated with the same type of vaccine. However, patients who received another type of influenza vaccine might have been included. Thirdly, we did not evaluate whether the occurrence of Bell’s palsy after influenza vaccination in the preceding season affects the risk of Bell’s palsy after vaccination in the following flu season. We identified that only two patients who received influenza vaccines developed Bell’s palsy in the risk interval (1–42 days postvaccination) for two of the three flu seasons, which was not a large number of patients. However, further studies using data from a longer period than this study may be needed to assess whether patients who developed Bell’s palsy after influenza vaccination in the previous season have an increased risk of Bell’s palsy in subsequent flu seasons. Finally, the safety profile may vary if the influenza vaccine is simultaneously delivered with other vaccines such as shingles in the elderly. We could not consider simultaneous vaccination because the data we used in the study did not contain any information related to vaccinations other than the influenza vaccine.

## 5. Conclusions

We found no evidence of an elevated risk of Bell’s palsy after influenza vaccination in Korean elderly. Although the risk of Bell’s palsy was high when the risk interval was set to be 1–28 days postvaccination in the 2017/2018 season, the consistency of null results at different control intervals indicated that our results were robust. The results of our study provided reassurance about the safety of influenza vaccine in the NIP program. However, it is necessary to continuously monitor the risk of Bell’s palsy during future flu seasons, given that the influenza vaccine formulations change annually. In addition, further study is needed because the risk of Bell’s palsy might be high in patients with chronic diseases such as diabetes and dyslipidemia, as shown in some subgroup analyses of this study.

## Figures and Tables

**Figure 1 vaccines-09-00746-f001:**
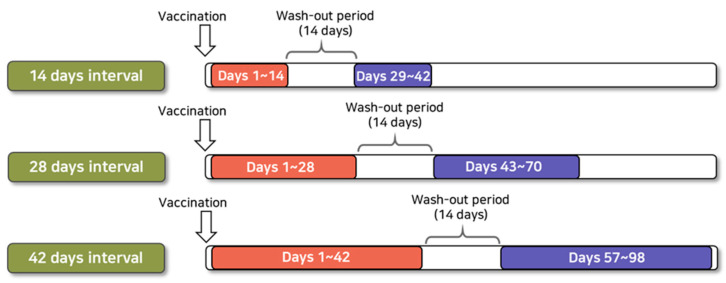
Illustration of risk and control intervals for self-controlled risk interval analysis.

**Figure 2 vaccines-09-00746-f002:**
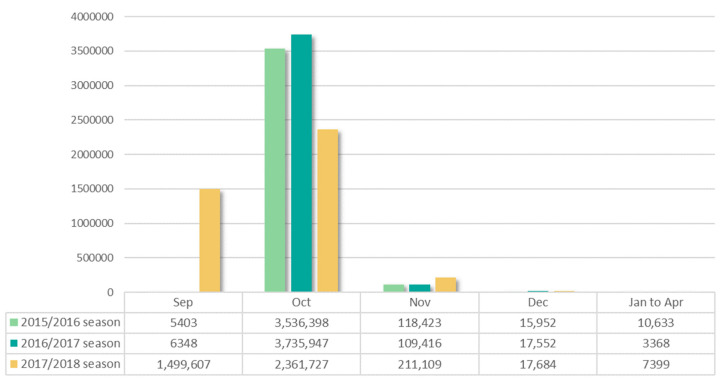
Distribution of influenza vaccination in the elderly during three flu seasons, 2015/2016 season through 2017/2018 season.

**Figure 3 vaccines-09-00746-f003:**
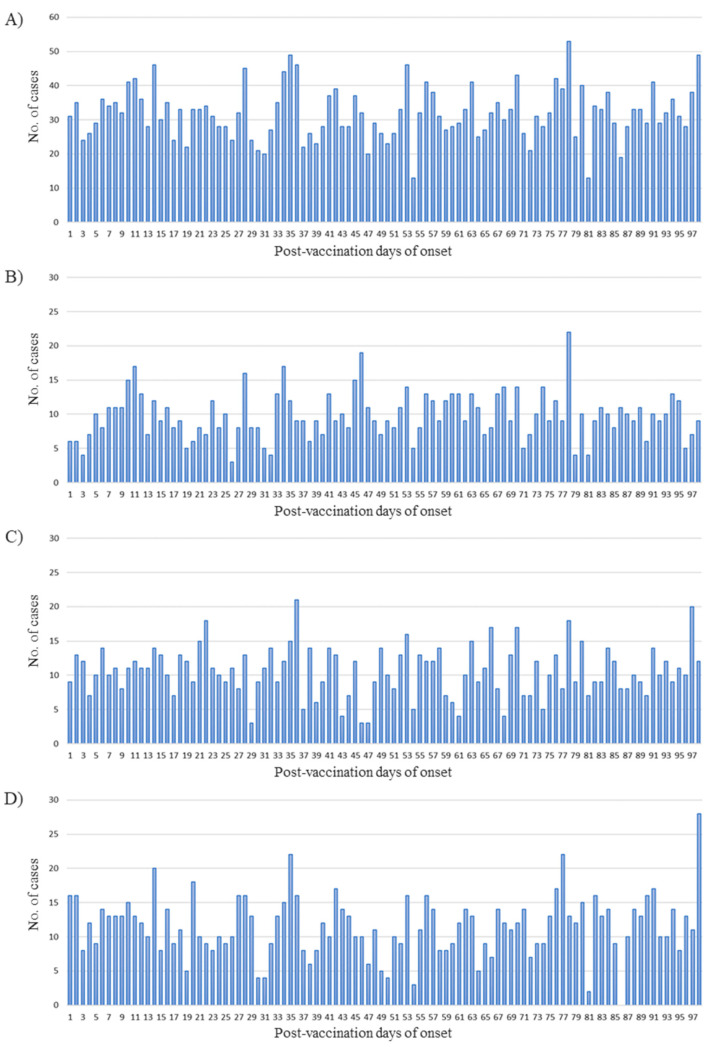
Distribution of the occurrence of Bell’s palsy 1–98 days following influenza vaccination, 2015/2016 season through 2017/2018 season. The graphs illustrate timing during (**A**) the 2015/2016 through 2017/2018 seasons, (**B**) the 2015/2016 season, (**C**) the 2016/2017 season, (**D**) the 2017/2018 season.

**Figure 4 vaccines-09-00746-f004:**
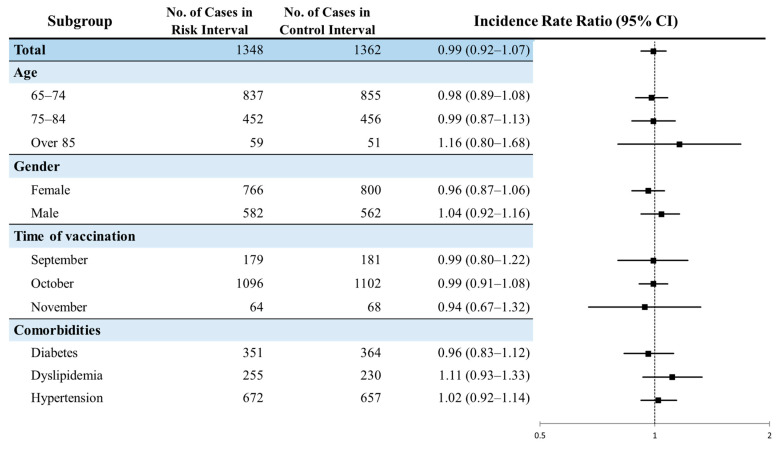
Subgroup analysis for the risk of Bell’s palsy after influenza vaccination.

**Table 1 vaccines-09-00746-t001:** Descriptive characteristics related to first doses of influenza vaccine in the elderly during the three flu seasons, 2015/2016 season through 2017/2018 season.

Characteristics	Total	2015/2016 Season	2016/2017 Season	2017/2018 Season
**Total**	11,656,966 (100%)	3,686,809 (100%)	3,872,631 (100%)	4,097,526 (100%)
Age				
65–74	6,556,985 (56.25%)	2,146,251 (58.21%)	2,170,690 (56.05%)	2,240,044 (54.67%)
75–84	4,167,349 (35.75%)	1,268,047 (34.39%)	1,390,244 (35.90%)	1,509,058 (36.83%)
Over 85	932,632 (8.00%)	272,511 (7.39%)	311,697 (8.05%)	348,424 (8.50%)
**Gender**				
Female	6,854,656 (58.90%)	1,515,265 (41.01%)	1,588,301 (41.46%)	1,698,744 (41.20%)
Male	4,802,310 (41.10%)	2,171,544 (58.99%)	2,284,330 (58.54%)	2,398,782 (58.80%)
**Time of vaccination**				
September	1,511,358 (12.97%)	5403 (0.14%)	6348 (0.16%)	1,499,607 (36.60%)
October	9,634,072 (82.65%)	3,536,398 (95.92%)	3,735,947 (96.47%)	2,361,727 (57.64%)
November	438,948 (3.77%)	118,423 (3.21%)	109,416 (2.83%)	211,109 (5.16%)
December	51,188 (4.39%)	15,952 (0.43%)	17,552 (0.45%)	17,684 (0.43%)
January	16,644 (1.43%)	7692 (0.21%)	2561 (0.07%)	6391 (0.16%)
February	3842 (0.03%)	2437 (0.07%)	609 (0.02%)	796 (0.02%)
March	748 (0.01%)	424 (0.01%)	159 (0.00%)	165 (0.00%)
April	166 (0.00%)	80 (0.00%)	39 (0.00%)	47 (0.00%)
**Comorbidities ***				
Diabetes	2,865,260 (24.58%)	919,029 (24.93%)	955,542 (24.67%)	990,689 (24.18%)
Dyslipidemia	3,147,132 (27.00%)	986,375 (26.75%)	1,051,388 (27.15%)	1,109,369 (27.07%)
Hypertension	6,481,603 (55.60%)	2,081,761 (56.47%)	2,163,660 (55.87%)	2,236,182 (54.57%)

* Included patients diagnosed with the disease at least twice in the 6 months prior to vaccination.

**Table 2 vaccines-09-00746-t002:** Baseline characteristics of Bell’s palsy cases after influenza vaccination.

Characteristics	Cases in Risk Interval		Cases in Control Interval		*p*
	Days 1–42 PostVaccination		Days 57–98 PostVaccination		
	N	(%)	N	(%)	
**Total**	1348	(100%)	1362	(100%)	
**Age at vaccination**					0.6982
65–74	837	(62.09%)	855	(62.78%)	
75–84	452	(33.53%)	456	(33.48%)	
Over 85	59	(4.38%)	51	(3.74%)	
**Gender**					0.3136
Female	766	(56.82%)	800	(58.74%)	
Male	582	(43.18%)	562	(41.26%)	
**Months of vaccination**					0.5153
January–February	1	(0.07%)	4	(0.29%)	
March–April	0	(0%)	1	(0.07%)	
September–October	1275	(94.58%)	1283	(94.20%)	
November–December	72	(5.34%)	74	(5.43%)	
**Type of vaccinated institution**					0.3853
Private health institution	1136	(84.27%)	1131	(83.04%)	
Public health institution	212	(15.73%)	231	(16.96%)	
**Vaccinated regions**					0.7946
Capital region	518	(38.43%)	526	(38.62%)	
Metropolitan city	344	(25.52%)	333	(24.45%)	
Others	486	(36.05%)	503	(36.93%)	
**Setting of diagnosis**					0.9288
Inpatient	295	(21.88%)	300	(22.03%)	
Outpatient	1053	(78.12%)	1062	(77.97%)	

**Table 3 vaccines-09-00746-t003:** Risk of Bell’s palsy after influenza vaccination.

Time Period	No. of Cases in Risk Interval	No. of Cases in Control Interval	IRR (95% CI)
Total	1348	1362	0.99 (0.92–1.07)
2015/2016 season	387	423	0.91 (0.80–1.05)
2016/2017 season	467	442	1.06 (0.93–1.20)
2017/2018 season	494	497	0.99 (0.88–1.13)

Abbreviations: IRR, incidence rate ratio; CI, confidence interval.

**Table 4 vaccines-09-00746-t004:** Sensitivity analyses for risk of Bell’s palsy after influenza vaccination.

Time Period	No. of Cases in Risk Interval	No. of Cases in Control Interval	IRR (95% CI)
Risk interval: Days 1–14, Control interval: Days 29–42			
Total	475	441	1.08 (0.95–1.23)
2015/2016 season	138	129	1.07 (0.84–1.36)
2016/2017 season	153	155	0.99 (0.79–1.23)
2017/2018 season	184	157	1.17 (0.95–1.45)
Risk interval: Days 1–28, Control interval: Days 43–70			
Total	907	866	1.05 (0.95–1.15)
2015/2016 season	258	304	0.85 (0.72–1.00)
2016/2017 season	312	276	1.13 (0.96–1.33)
2017/2018 season	337	286	1.18 (1.01–1.38)

Abbreviations: IRR, incidence rate ratio; CI, confidence interval.

## Data Availability

This research database was built for the government R&D project, and access and utilization for other purposes are limited, so data sharing is not applicable.

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
