# Peer review of "Association between Influenza Vaccination and the Risk of Bell’s Palsy in the Korean Elderly"

_vaccines, 2021, doi:10.3390/vaccines9070746_

Round 1
Reviewer 1 Report
This is a well-designed study which evaluated the relation between influenza vaccination and the risk of Bell’s palsy. I have some question for this study.
Do you have the data of the reliability of the selection of Bell’s palsy using the ICD-10 codes and oral steroid medication?
Why did you follow up the participant only the short limited time? Is there any reference for the setting of this short periods of the evaluation? I mean if the author selected the longer periods of time for the patients (follow up and interval), the result could be the different. As the prevalence of Bell’s palsy was very low, the possibility of the development of Bell’s palsy during follow up periods would be very low, even though the authors evaluated very large participants.
How did you select the diabetes, dyslipidemia, and hypertension?
Reviewer 2 Report
The objectives and design of the study are well described. The description should be benefit by analyzing separately if the number of vaccines received during the development of the study should influence in the risk of developing facial palsy
Reviewer 3 Report
Dear Authors
I find the chosen topic very interesting and relevant. Moreover, the sample used is very large and representative. In order to accept it you will have to make some minor modifications.
1. Material and methods
a. Why did you only consider the elderly who received a single dose of vaccine in the three seasons? Wouldn't it be interesting to see if the risk increases as the doses increase? Justify
2. Results
a. Why didn't you calculate CIs for such large samples? I recommend that you calculate the percentages and corresponding CIs and include them in table 2, 3 and 4 and in figure 4,
